# Cartilage Collagen Neoepitope C2C Expression in the Articular Cartilage and Its Relation to Joint Tissue Damage in Patients with Knee Osteoarthritis

**DOI:** 10.3390/biomedicines12051063

**Published:** 2024-05-11

**Authors:** Taavi Torga, Siim Suutre, Kalle Kisand, Marina Aunapuu, Andres Arend

**Affiliations:** 1Department of Anatomy, University of Tartu, Ravila 19, 50411 Tartu, Estonia; siim.suutre@ut.ee (S.S.); marina.aunapuu@ut.ee (M.A.); andres.arend@ut.ee (A.A.); 2Department of Internal Medicine, University of Tartu, L. Puusepa 8, 50406 Tartu, Estonia; kalle.kisand@ut.ee

**Keywords:** osteoarthritis, collagen neoepitope C2C, cartilage, immunohistochemistry

## Abstract

Pathological cleavage of type II collagen (Col2) and generation of Col2 neoepitopes can serve as useful molecular markers of the progression of osteoarthritis (OA). One of such potential biomarkers is type II collagen neoepitope C2C. The aim of this study was to correlate the degree of articular cartilage damage in OA patients with C2C expression in histological samples of tissues removed during total knee replacement. Cartilage samples were obtained from 27 patients ranging in age from 55 to 66 years. In each patient, medial and lateral tibia plateau samples were analyzed according to the OARSI histopathology grading system. The C2C expression was evaluated on histological slides by semi-quantitative analysis using ImageJ Fiji 2.14.0 software. Spearman’s rank correlation analysis revealed a positive weak correlation (rho = 0.289, *p* = 0.0356) between the histological grade of tissue damage and the percentage of C2C staining. In addition, a highly significant positive correlation (rho = 0.388, *p* = 0.0041) was discovered between the osteoarthritis score (combining the histological grade of damage with the OA macroscopic stage) and the percentage of C2C staining in the samples. The C2C expression was detected in all the regions of the articular cartilage (i.e., the superficial zone, mid zone, deep zone and tidemark area, and the zone of calcified cartilage). Our findings imply that local expression of C2C correlates with the articular cartilage damage in OA-affected knees. This confirms that C2C can be used as a prospective marker for assessing pathological changes in the OA course and OA clinical trials.

## 1. Introduction

Osteoarthritis (OA) is the most prevalent form of arthritis, resulting in joint pain and disability [1]. This disease is related to the ageing population and obesity and predominantly affects weight-bearing joints [2]. OA mainly affects the hips, knees and spine but also hands and feet [3]. The main symptoms of OA are pain, stiffness and dysfunction of joints, which worsen the overall quality of life [4]. Although cartilage loss with the biomechanical changes in cartilage and chondrocytes is the main pathological feature of OA, the whole joint, including the subchondral bone, synovium, infrapatellar fat pad, menisci and ligaments, is affected [5]. It is suggested that factors contributing to the development of OA include advanced age, being overweight or obese, a history of injury or surgery to a joint, overuse from repetitive movements of the joint, misalignment of joints and a family history (genetics) of OA [6]. While there are currently no disease-modifying OA drugs (DMOADs) on the market, a number of clinical trials are being conducted based on the known pathophysiological processes of OA [7,8]. In recent years, attention has begun to be paid to OA-associated molecular biomarkers from biofluids to identify early pathological molecular shifts, which are not detected with traditional imaging methodology [9]. Such shifts may be indicative of the deterioration of joint health before any other signs, such as pain or evidence from imaging studies, appear. If proven to be sensitive enough, biomarkers may help to describe the course of the disease and provide evidence if the treatment modalities, such as physiotherapy, are working.

The articular cartilage consists of chondrocytes and the extracellular matrix (ECM). The ECM is composed of two main protein components: type II collagen (Col2) and aggrecan, the sulfated proteoglycan [10]. The Col2 molecule is the fundamental structural component of articular cartilage. It is composed of three identical α polypeptide chains of 1060 amino acid residues, arranged in a large triple helix and comparatively brief nonhelical telopeptides consisting of 19 amino acid residues in N-telopeptide and 27 amino acid residues in C-telopeptide [11,12]. To maintain healthy cartilage, there is a delicate equilibrium of extracellular matrix remodeling: a balance between matrix synthesis and matrix degradation [13]. Cartilage breakdown without a sufficient compensatory synthesis of new tissue leads to clinical features of OA, such as misalignment of the joint, inhibited mobility and pain [14]. Therefore, the focus should be on cartilage ECM-associated molecules, in particular, Col2 degradation molecules. Many biochemical markers associated with Col2 that reflect cartilage turnover have been proposed as candidate prognostic markers of the rapid progression of OA [15,16,17].

In the early stages of OA, the molecular composition and organization of the ECM is altered, but the surface of the cartilage remains intact [18]. Col2 undergoes degradation facilitated by proteolytic enzymes, including matrix metalloproteinases (MMPs) [19]. MMPs, such as collagenases MMP-1, MMP-8 and MMP-13, exhibit the capability to cleave the native triple-helix structure of collagen. As a result of the cleaving, two new peptides, three-quarter-length and one-quarter-length mature Col2, are formed with specific neoepitopes at cleavage sites [20,21]. Such neoepitopes are, for example, C-telopeptide of type II collagen (CTX-II), COL2-3/4C Short (C1,2C), C2M and type-II collagen C-terminal cleavage neoepitope (C2C). CTX-II presence has been identified in serum and urine, where it is referred to as sCTX-II and uCTX-II, respectively [22,23]. Furthermore, uCTX-II, which has been demonstrated to associate with OA progression, is currently one of the most evaluated OA biomarkers [24]. CTX-II has been shown to be present not only in the damaged articular cartilage, but also in the tidemark and calcified cartilage at the interface with bone [25], and, moreover, CTX-II is supposed to be a marker of subchondral bone degeneration [16]. C1,2C levels have been shown to increase in OA cartilage compared to healthy samples; however, it has been observed that the accuracy of the C1,2C assay is compromised by its tendency to cross-react with type I collagen, primarily indicative of bone turnover [20]. Increased uC2C levels have been correlated with knee pain and with declined functional abilities of the lower limb [26]. The few other studies have reported increased levels of uC2C in subjects with radiographically defined knee OA [27,28], and uC2C has been found to associate with knee OA progression [29]. However, in a study by Boeth et al. tracing OA progression, no significant difference in the urine content of C2C was found [30]. Conrozier et al. compared serum concentrations of C2C in patients with multiple-site OA to patients with only hip OA and found that C2C levels were higher when the patient was suffering from just hip OA [31].

It is important to highlight that there has been relatively little histopathological research specifically looking at neoepitopes, such as C2C, in the cartilage. There are only a few studies that stand out in this area. Research indicates that C2C may be detectable within cartilage, exhibiting immunohistochemical staining patterns similar to the ones observed with the C1,2C antibody [11,32]. These studies introduced the methods to explore different epitopes of collagen that had been degraded by enzymes. This illustrates that additional biomarkers in the characterization of OA are still to be sought, and, therefore, there is notable interest in the identification and research of the behavior of C2C, which has been detected at elevated levels in OA cartilage and body fluids (serum, urine), presenting a potential laboratory metric for assessing cartilage breakdown [33,34].

Although recent studies have focused on C2C detection in urine by associating C2C and knee OA (kOA) radiographic features [27,35,36], our study is aimed at histological analysis of C2C in cartilage samples. Given the Human Protein Atlas’ identification of Col2 in diverse tissues, including connective tissues, gastrointestinal tissues, endocrine tissues and eyes, the concern persists that the source of excreted C2C may not exclusively be cartilage. Consequently, the potential use of C2C as an OA biomarker remains uncertain. The aim of this study was to detect the local expression levels of C2C in the articular cartilage of kOA removed during total knee replacement (TKR) and to determine if the levels of C2C expression correlate with the cartilage histological damage and the score of OA for the affected knee.

## 2. Materials and Methods

### 2.1. Sampling

Cartilage tissue was obtained from 27 patients undergoing primary unilateral TKR due to end-stage kOA at the Traumatology and Orthopedics Clinic of the University of Tartu (Estonia) in 2017–2018. The inclusion criterion was radiologically diagnosed end-stage kOA (Kellgren/Lawrence grade 3–4) before the age of 70 years. Seventeen women and ten men aged from 55 to 66 years were enrolled and examined in the study. Patients who showed indications of acute infection within the past three months were excluded. Other exclusion criteria included secondary OA caused by trauma, gout, infections, or congenital and developmental problems affecting the knee joints, as well as rheumatoid arthritis.

Written informed consent was received from all the patients before participation. The research project was approved by the Ethics Review Committee on Human Research of the University of Tartu (Protocol no. 265T-22, 19 December 2016), and it is in accordance with the Declaration of Helsinki [37]. Tissue punches included cartilage layers and the subchondral bone. In each subject, cylindrical osteochondral explants of 7 mm diameter were drilled perpendicular to the articular surface immediately after TKR from two load-bearing sites of the tibial plateau—one designated as meniscus-covered medial (MTM) and the other as meniscus-covered lateral (LTM) tibial plateau. In one case, the sample from the meniscus-covered medial tibial plateau (MTM) was lost due to technical problems (see Appendix A).

### 2.2. Histological Analysis

For histopathological assessment, MTM and LTM samples were fixed in 10% buffered formalin solution. Decalcification was performed using Sakura TDE^TM^ 30 Decalcifier System (Sakura Finetek Europe, Alphen aan den Rijn, The Netherlands) with Sakura Decalcifier Solution (Sakura Finetek Europe, Alphen aan den Rijn, The Netherlands) during 24 h and embedded in paraffin with a vacuum infiltration processor Sakura Tissue-Tek^®^ VIP 5 Jr, (Sakura Finetek Japan, Tokyo, Japan). Sections 7 µm thick were cut with the microtome Microm Ergostar HM200 (Microm GmbH, Walldorf, Germany), mounted on AutoFrost^®^ Microscope Slides (Cancer Diagnostics, Inc., Durham, NC, USA), dried overnight on a slide dryer heating plate at the temperature of 37 °C, deparaffinized with xylene and rehydrated in a series of graded ethanol solutions. Then, the sections were stained using the Safranin O (Sigma-Aldrich Chemie GmbH, Taufkirchen, Germany) staining method, and finally, the sections were dehydrated in graded ethanol solutions, in xylene, and mounted with Eukitt^®^ medium (Sigma-Aldrich Chemie GmbH, Taufkirchen, Germany) before applying the coverslips. We used the OARSI OA Cartilage Histopathology Assessment System for evaluation of the level of local cartilage pathology, ranging from grade 0 (normal cartilage architecture) to grade 6 (severe damage) [38]. Grade 0 is characterized by a normal matrix architecture and smooth surface; cells are intact and in appropriate orientation. For grade 1, only the superficial zone is affected. The surface of the cartilage is generally intact but may show superficial fibrillation. The middle and deep zones are not affected. Grade 2 is characterized by discontinuity of the cartilage superficial zone by focal fibrillation. Small portions of cartilage from the superficial zone are detached due to abrasion from shear forces. Grade 3 is characterized by vertical fissures extending from the superficial zone into the middle zone or even into the deep zone. The grade 4 key feature is erosion. If, in the earliest stage, cartilage matrix loss leads to delamination of the superficial zone, then more extensive erosion causes excavation and loss of matrix in fissured domains. The grade 5 key feature is denudation. Unmineralized cartilage is completely eroded to the level of mineralized cartilage or bone. Grade 6 is recognized by deformation. The contour of the articular cartilage is changed by processes of microfracture, repair and bone remodeling. The macroscopic stage of OA was assessed by visually evaluating the tibial plateau removed during the operation. To evaluate the extent of articular surface involved with OA, the staging method ranging from stage 1 (<10% of involvement of the articular cartilage area) to stage 4 (>50% of involvement) was used. For OA scoring, a simple formula was used: OA score = OA grade × OA stage [38]. Two independent assessors performed the evaluation of the articular cartilage damage. In the case of the few evaluation disagreements, the specimens were revised to formulate the final assessment.

### 2.3. Immunohistochemistry (IHC)

For immunohistochemical analysis, 4 µm thick paraffin-embedded sections were cut using the microtome Microm Ergostar HM200 (Microm GmbH, Walldorf, Germany), mounted on AutoFrost^®^ Microscope Slides (Cancer Diagnostics, Inc., Durham, NC, USA) and dried overnight on a slide dryer heating plate at the temperature of 37 °C. The sections were deparaffinized in xylene and rehydrated in a series of graded ethanol solutions. To inactivate endogenous peroxidase activity, the sections were treated for 10 min with freshly prepared 0.6% H_2_O_2_ (Honeywell Fluka^TM^, Seelze, Germany) dissolved in distilled water. Then, the sections were washed for 3 × 5 min in 1× GibcoTM Phospate Buffered Saline (PBS, pH 7.4) solution (Thermo Fisher Scientific, Landsmeer, The Netherlands). After washing, the sections were incubated with 100 µL Dako Antibody Diluent (cat no S2022) (Dako Denmark A/S, Glostrup, Denmark) to block nonspecific antibody binding. After blocking, the sections were incubated with 100 µL C2C monoclonal antibody (product no 50-1018; Ab 5109; produced by IBEX Pharmaceutical Inc., Mount Royal, QC, Canada) in the dilution 1:10,000 for one hour at room temperature. This antibody is produced in mouse cells, used in the Type II Collagen Cleavage Sandwich Assay developed by IBEX Pharmaceuticals Inc. and is suitable for detecting the C2C epitope in human samples. The antibody was diluted in Dako Antibody Diluent (cat no S2022) (Dako Denmark A/S, Glostrup, Denmark). Thereafter, to visualize the primary antibody, the commercial kit Dako RealTM EnvisionTM Detection System, Peroxidase/DAB+, Rabbit/Mouse (cat no K5007) (Dako Denmark A/S, Glostrup, Denmark) was used. Washing 3 × 5 min between each step after incubation with the primary antibody was performed in 1× GibcoTM Phospate Buffered Saline (PBS, pH 7.4) solution (Thermo Fisher Scientific, Landsmeer, The Netherlands) containing 0.07% Tween 20 detergent (BioTop, Naxo, Tartu, Estonia). All incubations were performed at room temperature in a humidified chamber. No positive IHC-staining was noted in negative control slides where the primary antibody was omitted. The sections were stained for 1 min with Toluidine-blue to counterstain the cartilage proteoglycans. Finally, the sections were dehydrated in a series of graded ethanol solutions, in xylene and mounted with Eukitt^®^ medium (Sigma-Aldrich GmbH, Taufkirchen, Germany) before applying the coverslips.

The tissue slides were fully scanned using a Leica SCN400 (Leica Microsystems, Wetzlar, Germany) with magnification 20×. The proportion of the stained tissue sample was assessed on the scanned images in a semi-quantitative manner with ImageJ 2.14.0 software and the color threshold function, as described by Crove and Yue [39]. In brief, a scanned IHC image was opened with the Aperio Image Scope program. The tissue section was selected using the Zoom navigation tool, and the selected area was snapshot and saved as a TIFF file. Consequently, the TIFF file was opened in the ImageJ (Fiji) program, and the “Color threshold” tool was applied to select the whole stained tissue section area. The selected area was analyzed by using the “Measure” tool. The color threshold was adjusted to the maximum to remove the background signal, without removing the DAB signal area. The results of the DAB signal area in the form of pixels were entered into an MS Excel table, and eventually, the percentage of the DAB staining area was calculated. In essence, the ImageJ software facilitated the assessment of pixel numbers within the tissue section, and the color threshold function enabled the selection and enumeration of pixels corresponding to positively stained regions with the C2C antibody, resulting in the percentage of the positively stained tissue section area for each sample (see Table 1). In total, 53 sections of samples were analyzed (due to technical problems, no MTM sample was obtained from one patient).

### 2.4. Statistical Analysis

Correlations between the extent of local expression of C2C in the articular cartilage and the degree of cartilage damage (both OARSI histopathology grade and OARSI score) were analyzed using Spearman’s rank non-parametric correlation analysis. The Wilcoxon signed-rank test was used to compare the OARSI histological grade, stage and C2C staining of LTM and MTM. Statistical analysis was performed, applying the GraphPad InStat software version 3.10 (GraphPad Software, San Diego, CA, USA). The level of significance was set at *p* < 0.05.

## 3. Results

The main findings of this study, such as the histopathology grade of the tissue damage, the stage of macroscopic damage of the tibial plateau, the kOA score and the proportion of the tissue sample stained positively for C2C, are summarized in Table 1, while patients’ individual data are provided in the Appendix A.

In each patient, two tissue samples were analyzed, which were obtained from different load-bearing sites of the knee joint: one from the meniscus-covered medial tibial plateau (MTM) and the other from the meniscus-covered lateral tibial plateau (LTM). As seen in Table 1, kOA histopathology grades (according to OARSI indicators referred to in the methodology section) varied from 0 to 6.5. Comparing the same patient’s LTM and MTM samples, the histopathology grades differed in most cases; equal grades were recorded in three patients only. Higher histopathology grades were mostly recorded for the medial side of the joint—in 17 cases, MTM samples had higher grades, while only in 6 cases, LTM samples had higher grades as compared to the counter side. The MTM OARSI histopathology grades and scores were statistically substantially higher than on the LTM side (*p* = 0.021 and *p* = 0.010, respectively, Wilcoxon signed-rank test). Furthermore, the percentage of C2C staining on the MTM side was larger than that on the LTM side (*p* = 0.022, Wilcoxon signed-rank test) (Table 1).

In Figure 1, representative micrographs are presented to depict histopathology grades 0, 1.5, 3.5 and 6. Moreover, we found a positive weak correlation of the percentage of C2C immunohistochemical expression with the OARSI histopathology grade (Spearman rho = 0.289 (corrected for ties); 95%CI: 0.0124 to 0.525; *p* = 0.0356; Figure 1, Table 1). However, this correlation did not remain significant when calculated only for females (n = 17) (Spearman rho = 0.189 (corrected for ties); 95%CI: −0.170 to 0.503; *p* = 0.285) or for males (n = 10) (Spearman rho = 0.369 (corrected for ties); 95%CI: −0.117 to 0.712; *p* = 0.121).

Furthermore, significant positive correlation was found between the OA score (combining the histological grade of damage with the OA macroscopic stage) and the percentage of C2C staining in the samples (Spearman rho = 0.388 (corrected for ties); 95%CI: 0.123 to 0.601; *p* = 0.0041). This correlation remained positive among the female participants (Spearman rho = 0.349 (corrected for ties); 95%CI: 0.002 to 0.621; *p* = 0.043), but not in the male subgroup (Spearman rho = 0.382 (corrected for ties); 95%CI: −0.102 to 0.720; *p* = 0.107). The macroscopic stage of OA was visually assessed on the tibial plateau removed during surgery (see Section 2). Our findings suggest indeed that the proportion of tissue stained for C2C may be related to the cartilage damage (see Figure 1).

Additionally, it may be pointed out that the OARSI histopathology grade and OA score were correlated with patients’ ages (Spearman rho = 0.293 (corrected for ties); 95%CI: 0.016 to 0.528; *p* = 0.033 and Spearman rho = 0.358 (corrected for ties); 95%CI: 0.089 to 0.578; *p* = 0.0085, respectively), but the percentage of C2C staining in the samples was not (Spearman rho = 0.169 (corrected for ties); 95%CI: −0.115 to 0.427; *p* = 0.227).

There was no correlation between the proportions of tissue staining between LTM and MTM samples (Spearman rho = 0.039 (corrected for ties); 95%CI: −0.364 to 0.431; *p* = 0.84). It could be due to the fact that more profound damage and, therefore, more intense staining are usually on one (lateral or medial) side of the tibial plateau.

No correlation was found between the extent of C2C staining and the Kellgren/Lawrence score, as well as with the KOOS score (the Estonian translation of the Knee Injury and Osteoarthritis Outcome Score questionnaire was used to assess the knee joint complaints of the respondents who were recruited).

## 4. Discussion

While the evaluation of OA traditionally relies on clinical manifestations, including pain, joint mobility loss and functional impairment, and specific radiographic parameters, there exists a need for a sensitive molecular marker that delineates the progression of the disease. A preferable biomarker would ideally be detectable from a readily accessible biological sample, such as urine or blood serum. Such a marker would facilitate a more intricate monitoring of patients, particularly those belonging to high-risk categories, such as individuals exhibiting poor responses to conservative or surgical interventions, those manifesting OA at a relatively young age (below 70) [40] or participants in clinical trials assessing OA treatments.

In the selection of a good disease marker, the objective is to prioritize stability and reliability. When evaluating C2C as a potential marker, it is essential to demonstrate the pathological alterations of the protein in histological samples. This approach provides the strongest evidence for establishing a correlation between the actual tissue damage and the levels of the biomarker in the cartilage. In further studies, the correlation among tissue damage, local biomarker levels and biomarker presence in easily accessible samples, such as blood or urine, should be established. One antibody specific to C2C is used in the Type II Collagen Cleavage Sandwich Assay developed by IBEX Pharmaceuticals Inc. Through collaboration with IBEX, we procured the antibody employed in the sandwich assay for our immunohistochemical experiments. IBEX did not provide any funding for this study, but was kind enough to supply the C2C-specific antibody on request. Notably, this antibody is not known to have been used in immunohistochemical analyses previously. The antibody against this type of peptide has been validated in Western blot analysis [41]. The proper staining pattern of the cartilage tissue matrix, which was found with the antibody, and the fact that the proportion of tissue stained for C2C correlated with the tissue damage, provided evidence that the proper epitope was detected with the antibody in immunohistochemical experiments. Our results showed the positive immunohistochemical staining in all samples obtained during TKR, and the C2C expression was detected in all the regions of articular cartilage (i.e., the superficial zone, the mid zone, the deep zone, the tidemark area and the zone of calcified cartilage), but the subchondral bone was generally not stained.

Considering that other tissues in the body could contribute to the content of excreted C2C, it is important to investigate the expression and localization of this neoepitope directly in the damaged cartilage. This type of investigation is of significance not only for C2C, but also for other biomarkers to prove their pathogenic significance in the context of the disease [27,42]. It is worth emphasizing that, even though detected and measured in body fluids, the investigation and quantification of neoepitopes in articular cartilage have been the subject of limited attention within the existing literature, resulting in a restricted pool of studies available for consideration. Correlation between CTX-II staining and tissue damage has been described in patients with femoral head necrosis [43]. COL2-3/4C Short (C1,2C) has been detected in the articular cartilage of mice, while C2M has been identified in the articular cartilage of humans. However, there is limited evidence regarding alterations in their expression and their relationship to tissue damage [44,45]. Poole et al. introduced an antibody designed to target C2C, but their study did not undertake an exploration of potential immunohistochemical applications, but instead focused on the quantification of the C2C in patients’ serum and urine samples using the ELISA test [41]. In a paper by Dejica et al., C2C and C1,2C were detected and described in articular cartilage, but the results of immunohistochemistry were not quantified [11]. An additional neoepitope of the C-terminus of the type II collagen TC fragment was assessed in equine articular cartilage, but the staining patterns were characterized solely by their intensity, lacking direct quantification [32]. Thus, there has been limited investigation into quantifying the neoepitopes at the tissue structural level, and, therefore, it would be of great interest to describe the local immunohistochemical expression pattern of C2C in articular cartilage.

In our study, we show that it is possible to detect C2C in articular osteoarthritic cartilage, and its expression patterns may correspond to the progression of OA. As a strength of the study, it can be said that the evaluation of immunohistochemical staining of C2C was performed with computer assistance, reducing the subjectivity of the estimations. On the other hand, the limitations of the study are that the antibody used was not originally intended for immunohistochemistry and the fact that the present study was carried out on a relatively small number of patients. Also, unfortunately, we could not obtain cartilage specimens from a normal cadaveric knee joint for comparison because of complicated legislation in Estonia. Nevertheless, we successfully demonstrated a correlation between C2C expression and both the grade of cartilage damage and the overall knee joint damage score as evaluated by OARSI indicators. This implies that C2C may possess indicative properties for the development of kOA. It is important to note that an ideal biomarker should demonstrate a strong correlation with the advancement of tissue damage, both at the tissue level and in more readily accessible biomaterials like plasma or urine. Subsequent investigations should ascertain whether local tissue alterations of C2C content are reflected in the C2C levels found in urine samples—an aspect explored to some extent by other researchers [46]. The identification of a reliable and readily assessable biomarker would not only facilitate more detailed patient monitoring, but also serve as a surrogate marker in the context of clinical trials.

In conclusion, our findings demonstrate that type II collagen neoepitope C2C is a useful molecular marker of OA. After analyzing histological samples from OA patients undergoing total knee replacement, we identified statistically highly significant positive correlations between local expression of C2C and both the histological degree of tissue destruction and the overall osteoarthritis score. Notably, C2C staining was detected in multiple locations of the articular cartilage. These findings highlight the potential value of C2C as a prospective marker for monitoring pathological changes in OA and guiding therapeutic trials to treat this debilitating ailment.

## Figures and Tables

**Figure 1 biomedicines-12-01063-f001:**
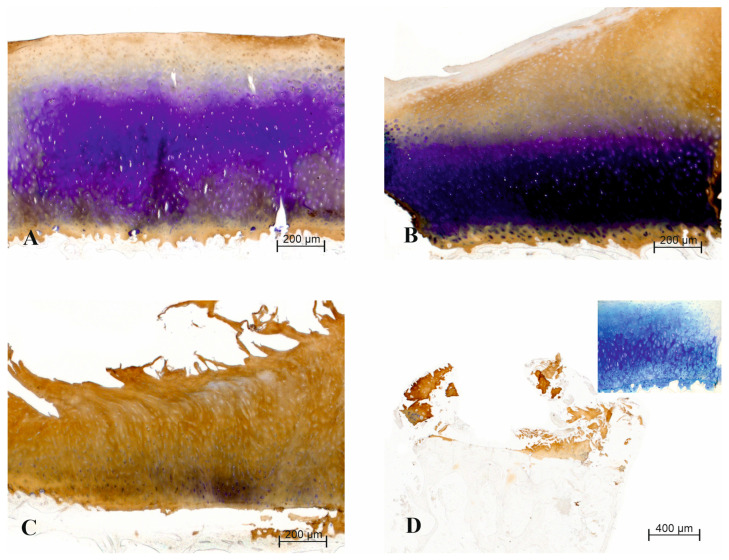
(**A**) Micrograph of a sample with histopathology grade 0: weak C2C immunostaining (brown) in the superficial and deep zones (counterstaining of proteoglycans with Toluidine-blue). The cartilage content of proteoglycans determined by the amount of Toluidine-blue staining was decreased in the tissue samples with more advanced OARSI tissue damage grade; (**B**) Micrograph of a sample with histopathology grade 1.5: moderate C2C immunostaining (brown) in the superficial and upper parts of the middle zone (counterstaining of proteoglycans with Toluidine-blue); (**C**) Micrograph of a sample with histopathology grade 3.5: overall abundant C2C immunostaining (brown), weak counterstaining in the deep zone (counterstaining of proteoglycans with Toluidine-blue); (**D**) Micrograph of a sample with histopathology grade 6: abundant C2C immunostaining (brown) in the remaining cartilage; subchondral bone, which occupies most of the specimen, is virtually not stained (counterstaining of proteoglycans with Toluidine-blue). A sample picture of negative control where the primary antibody was omitted is presented as an inset in the upper-right corner of micrograph (**D**). All micrographs were taken using a 5× objective.

**Table 1 biomedicines-12-01063-t001:** Results of histopathology assessments and C2C staining of samples from 27 patients (10 male and 17 female patients; in each subject, two load-bearing sites of the tibial plateau were analyzed).

Patients’ Age (in Years)	Tibial Plateau Site	OARSI Histo-Grade	OARSI Score (Grade × Stage)	Assessment of C2C Staining, %
59 (58–64) Min–max = 55–66	LTM + MTM (n = 53) *	3.0 (2.5–4.3) ^a^ Min–max = 0–6.5	8.0 (5.0–10.5) ^b^ Min–max = 0–26	73.2 (53.2–89.5) ^a,b^ Min–max = 16.2–100
	LTM (n = 27)	2.5 (2.0–3.5) ^c^ Min–max = 0–6	6.85 (4.6–10.4) ^d^ Min–max = 0–13.5	60.4 (49.0–84.6) ^e^ Min–max = 16.2–96.1
	MTM (n = 26) *	3.7 (2.6–4.5) ^c^ Min–max = 1.5–6.5	8.0 (5.3–13.5) ^d^ Min–max = 2–26	79.6 (68.1–94.2) ^e^ Min–max = 42.3–100

Data are presented as median with interquartile range in brackets; min–max—minimal and maximal values. LTM—lateral tibial plateau, meniscus-covered area biopsy; MTM—medial tibial plateau, meniscus-covered area biopsy. ^a^ Positive correlation of the percentage of C2C immunohistochemical expression with the OARSI histopathology grade (Spearman rho = 0.289, corrected for ties, *p* = 0.0356) and ^b^ with the OARSI score (Spearman rho = 0.388, corrected for ties, *p* = 0.0041). ^c^ Significant differences of OARSI histo-grades between LTM and MTM (the two-tailed *p* = 0.021; Wilcoxon signed-rank test), ^d^ significant differences of OARSI score between LTM and MTM (the two-tailed *p* = 0.010; Wilcoxon signed-rank test) and ^e^ significant difference of C2C staining between LTM and MTM (the two-tailed *p* = 0.022; Wilcoxon signed-rank test). * Due to technical problems, no MTM sample was obtained from one patient.

## Data Availability

Data are unavailable due to privacy restrictions.

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
