# Peer review of "Cartilage Collagen Neoepitope C2C Expression in the Articular Cartilage and Its Relation to Joint Tissue Damage in Patients with Knee Osteoarthritis"

_biomedicines, 2024, doi:10.3390/biomedicines12051063_

Round 1
Reviewer 1 Report
Comments and Suggestions for Authors
The authors found an association between C2C expression in the articular cartilage and the level of cartilage damage in OA patients.
Comments
1. Line 42-43: This statement is not true. The authors should modify it.
2. Line 58, 208: These sentences are not clear. They should be clarified.
3. Lines 64-93: This part of introduction is messy and contains many mistakes. The authors should learn the differences between collagenases and other matrix metalloproteinases and mechanisms of type II collagen breakdown. The authors should differentiate biomarkers which they describe and to show their potential ore clearly.
4. Introduction should include description of previous studies (PMID:31053993, PMID: 27481905, PMID: 19404937, PMID: 18578964) on C2C as biomarker for OA.
5. Lines 89-92: The importance of description of these studies is not clear. This should be clarified.
6. Line 59: Reference is required at the end of this sentence.
7. Lines 115-117: The number of the examined patients is not clear. This should be clarified.
8. Line175: The use of mouse-α-C2C monoclonal antibody for human samples is not clear. This should be clarified.
9. Line 204: The number of analyzed sections should be indicated.
10. Line 210: Software version should be indicated.
11. Table 1 should be moved to supplementary file. However, the data from Table 1 should be analyzed and presented accordingly. This should be corrected.
12. Figure 1: Magnification bars should be supplied for each fragment.
13. Limitations and strengths of the study should be added.
14. Conclusions are missing. This should be corrected.
Comments on the Quality of English Language1. Lines 58, 208: These sentences are not clear. They should be clarified.
Reviewer 2 Report
Comments and Suggestions for Authors
My major concern is related to the use of an antibody not validated for IHC supplied by IBEX. The role of IBEX in this manuscript was not declared.
Other comments:
Lines 33-35: references are lacking.
Lines 37-39: the authors forget to mention the infrapatellar fat pad. It should be added.
There is no mention of OA biomechanical changes of cartilage and chondrocytes. It should be added.
Line 78 :what about MMP3?
Lines 94-96: what did these papers find? Results should be reported.
Section 2.1: inclusion/esclusion criteria should be added.
Lines 210-211: graphpad software should be cited as reported here: https://www.graphpad.com/guides/prism/latest/user-guide/citing_graphpad_prism.htm
First part of the results should be moved to the discussion. The authors should show data in this section.
Line 234: this correlation is weak.
Did the authors calculate kellgren Lawrence score of these patients?
What about pain? Could the authors check if there is a correlation with pain (VAS) and C2C staining?
Could the authors compare LTM C2C staining with MTM C2C staining to see if there is a difference between the two regions?
Did the authors evaluate synovial/infrapatellar fat pad inflammation to see if there is a correlation with C2C staining?
In the discussion, the authors reported “… demonstrate the pathological alterations of the protein in histological samples. This approach provides the strongest evidence for establishing a correlation between the actual tissue damage and the levels of the biomarker in easily accessible samples such as blood or urine.” However, the authors did not evaluate nor correlate the levels of C2C in urine or blood.
In the discussion, it is mentioned that the authors received the antibody used in this paper Ab 5109 from IBEX Pharmaceutical Inc and that this is the first time that it is used. Indeed, I am not able to find this antibody on the website of the supplier. This is questionable. What is the role of IBEX in this study? Did the authors use a commercial antibody against C2C to confirm the data obtained?
Limitations of the study should be discussed.
The authors declare that no funding were received. So, how did the authors perform the IHC evaluation? Again, what is the role of IBEX?
Round 2
Reviewer 1 Report
Comments and Suggestions for Authors
The authors improved the quality of their manuscript. However, some comments were not addressed.
Comments
1. Previous Comment #11 was not addressed properly: Table 1 presents the raw data. It should be statistically explored. This should be corrected.
2. Lines 13, 15, 72, 294: Correct writing type II collagen. This should be corrected.
3. Lines 106-107: Prognostic significance of C2C in OA development would be very difficult to use in the medical setting as articular cartilage is available only at the end-stage patients with OA after the joint replacement. This sentence should be rephrased.
4. Line 241: The authors should indicate that correlation was positive and weak. This should be corrected.
5. Lines 242-245: The authors should indicate here the number of female and males, respectively.
6. Line 295: The authors cannot discuss the “early molecular markers” related to their study as C2C staining in cartilage is possible only at the end-stage patients. This should be corrected.
7. Lines 365-366: C2C cannot be useful for tracking the course of OA as C2C staining in cartilage is possible only at the end-stage patients. This should be corrected.
Reviewer 2 Report
Comments and Suggestions for Authors
No other comments
Author Response
Reviewer has no other comments.